# Effect of Citric Acid Cycle Genetic Variants and Their Interactions with Obesity, Physical Activity and Energy Intake on the Risk of Colorectal Cancer: Results from a Nested Case-Control Study in the UK Biobank

**DOI:** 10.3390/cancers12102939

**Published:** 2020-10-12

**Authors:** Sooyoung Cho, Nan Song, Ji-Yeob Choi, Aesun Shin

**Affiliations:** 1Department of Preventive Medicine, Seoul National University College of Medicine, Seoul 03080, Korea; ssooy7@snu.ac.kr; 2Cancer Research Institute, Seoul National University, Seoul 03080, Korea; Nan.Song@stjude.org (N.S.); miso77@snu.ac.kr (J.-Y.C.); 3Department of Epidemiology and Cancer Control, St. Jude Children’s Research Hospital, Memphis, TN 38105, USA; 4Department of Biomedical Sciences, Graduate School of Seoul National University, Seoul 03080, Korea; 5Medical Research Center, Institute of Health Policy and Management, Seoul National University, Seoul 03080, Korea

**Keywords:** colorectal neoplasms, citric acid cycle, single nucleotide polymorphism, obesity, physical activity, diet

## Abstract

**Simple Summary:**

The citric acid cycle has a central role in the cellular energy metabolism and biosynthesis of macromolecules in the mitochondrial matrix. We identified the single nucleotide polymorphisms (SNPs) of the citrate acid cycle with colorectal cancer susceptibility in UK population. Furthermore, we found the significant interaction of SNPs in the citric acid cycle with the contributors to energy balance and SNP-SNP interactions. Our findings provide clues to the etiology in cancer development related to energy metabolism and evidence on identification of the population at high risk of colorectal cancer.

**Abstract:**

Colorectal cancer is a common malignancy worldwide. Physical activity and a healthy diet contribute to energy balance and have been recommended for the prevention of colorectal cancer. We suggest that the individual differences in energy balance can be explained by genetic polymorphisms involved in mitochondria, which play a central role in energy metabolism at the cellular level. This study aimed to evaluate the association between genetic variants of the mitochondrial citric acid cycle and colorectal cancer. Study participants comprised 3523 colorectal cancer cases and 10,522 matched controls from the UK Biobank study. Odds ratios (ORs) and 95% confidence intervals (CIs) for colorectal cancer were estimated using a conditional logistic regression model. We found a significant association between the *SUCLG2* gene rs35494829 and colon cancer (ORs [95% CIs] per increment of the minor allele, 0.82 [0.74–0.92]). Statistical significance was observed in the interactions of the citric acid cycle variants with obesity, energy intake, and vigorous physical activity in colorectal cancer. We also identified significant SNP-SNP interactions among citric acid cycle SNPs in colorectal cancer. The results of this study may provide evidence for bioenergetics in the development of colorectal cancer and for establishing a precise prevention strategy.

## 1. Introduction

Colorectal cancer is commonly diagnosed worldwide. The GLOBOCAN estimated the incidence of colorectal cancer as the third most common cancer among men and the second most common cancer among women worldwide in 2018, with geographic and ethnic variation [1]. In Korea, colorectal cancer was the second most common cancer in 2017 [2]. The age-standardized incidence rate for colorectal cancer was 32.0 per 1 million, and it increased by 5.9% annually from 1999 to 2010 and decreased by 4.2% annually from 2010 to 2017. The age-standardized incidence rate for this cancer was higher among men (ASR, 38.8 per 1 million) than among women (ASR, 21.8 per 1 million).

The World Cancer Research Fund and American Institute of Cancer Research (WCRF/AICR) have reported lifestyle risk factors for colorectal cancer, such as alcohol consumption, obesity, and physical inactivity with “convincing” evidence and energy intake with “limited” evidence [3]. These lifestyle factors, including obesity, physical activity, and energy intake, are risk factors for colorectal cancer and a major factor in energy balance. Energy expenditure is categorized as resting energy expenditure and non-resting energy expenditure. Resting energy expenditure is the energy expenditure through minimal metabolism required to support essential body functions, and it is the largest component of energy expenditure [4]. Non-resting energy expenditure consists of exercise thermogenesis from exercise (physical activities), diet (ingesting, absorbing, metabolizing, and storing nutrients from food) and non-exercise activity (energy expended during non-exercise movements such as fidgeting or normal daily activities) [4]. Energy expenditure, including thermogenesis and basal metabolic rate, is closely associated with cell metabolism [5,6].

Mitochondria are highly appreciated as biosynthetic and bioenergetic organelles for their role in producing metabolites and ATP, which are byproducts of the citric acid cycle and the mitochondrial membrane potential, respectively. The citric acid cycle has a central role in the cellular energy metabolism and biosynthesis of macromolecules through a series of biochemical reactions occurring in the mitochondrial matrix. It is implicated that the abnormal function of the citric acid cycle can lead to pathological conditions. An in vitro study reported a significant association between the intermediates of the citric acid cycle and the regulation of hypoxia-inducible factor (HIF), which is a transcription factor for angiogenesis, glucose utilization, and apoptosis [7,8,9]. The activity of the citric acid cycle enzymes, including citric synthase, is reduced in mice in nutrient excess conditions [10]. Tumor cells separate processes from the citric acid cycle, allowing them to respond to elevated metabolic levels using additional energy sources such as glutamine, which was established as an essential nutrient source in various types of cancer [11].

We aimed to assess the polymorphism related to the mitochondrial citric acid cycle in colorectal cancer and examine the possible interaction between SNPs in the citric acid cycle and the contributors to energy balance, including obesity, physical activity, and energy intake. Furthermore, we suggested possible pairwise SNP interactions of the citric acid cycle on cancer due to the nature of the cycle. SNP-SNP interactions of pairwise SNPs in the citric acid cycle on colorectal cancer were also examined.

## 2. Results

### 2.1. Participant Characteristics

Table 1 shows the information on the selected SNPs in the study. Among the 24 selected SNPs, rs16832869 in the *SDHC* gene and rs2303436 in the *DLAT* gene were excluded from the SNP-SNP interaction analyses due to strong linkage disequilibrium with an r^2^ over 0.9 with rs16832884 in the *SDHC* gene (r^2^ = 0.998) and rs10891314 in the *DLAT* gene (r^2^ = 0.962), respectively. Finally, 22 SNPs in the citric acid cycle were included in the analyses.

Table 2 summarizes selected baseline characteristics of matched variables among cases and controls. A total of 10,522 controls and 3523 cases were included in the analyses. Participants aged 61–65 (*n* [%]; 1123 [31.9%] cases) were the most common, followed by 66–70 (1021 [29.0%] cases), 56–60 (412 [11.7%] cases), 51–55 (673 [19.0%] cases), 46–50 (193 [5.5%] cases), 41–45 (93 [2.6%] cases) and 36–40 (8 [0.2%] cases) at enrollment. There were more men (2024 [57.5%] cases) than women (1499 [42.5%] cases) among colorectal cancer cases. Most of the patients with colorectal cancer were White (3423 [97.2%] cases), followed by Asian or Asian British (31 [0.9%] cases), Black or Black British (28 [0.8%] cases), Mixed (18 [0.5%] cases), and Chinese (6 [0.2%] cases). In addition to the matching variable, the proportion of never smoker was higher among cases than among controls (number and proportion of never, previous and current smoker, respectively; 5315 [50.5], 4162 [39.6] and 1002 [9.5] among controls; 1607 [45.6], 1550 [44.0] and 350 [9.9] among cases).

Study participants were enrolled at 22 assessment centers in London (Barts, Croydon, and Hounslow), North East England (Middlesbrough and Newcastle), South East England (Oxford and Reading), North West England (Bury, Liverpool, and Manchester), South West England (Bristol), West (Stoke and Birmingham) and East (Nottingham) midlands of England, Yorkshire and the Humber (Leeds and Sheffield), Scotland (Edinburgh and Glasgow) and Wales (Swansea, Wrexham, and Cardiff). The most study participants were enrolled in West England (Bristol, 296 [8.4%] cases; Bury, 236 [6.7%] cases; Liverpool, 226 [6.4%] cases; Manchester, 114 [3.2%] cases; Stockport, 5 [0.1%] cases), followed by East England (Newcastle, 300 [8.5%] cases; Reading, 229 [6.5%] cases; Oxford, 128 [3.6%] cases; Middlesbrough, 118 [3.3%] cases), Midlands (Nottingham, 236 [6.7%] cases; Stoke, 154 [4.4%] cases; Birmingham, 135 [3.8%] in cases), Yorkshire and the Humber (Leeds, 309 [8.8%] cases; Sheffield, 193 [5.5%] cases), London (Hounslow, 157 [4.5%] cases; Croydon, 136 [3.9%] cases; Barts, 76 [2.2%] cases), Scotland (Glasgow, 158 [4.5%] cases; Edinburgh, 148 [4.2%] cases), and Wales (Cardiff, 152 [4.3%] cases; Swansea, 15 [0.4%] cases; Wrexham, 2 [0.1%] cases).

Participants whose Townsend deprivation index was −6.26 to −3.65 numbered 2788 (26.5%) in controls and 933 (26.5%) in cases those with −3.65 to −2.15 were 2733 (26.0%) in controls and 916 (26.0%) in cases; those with −2.15 to 0.515 were 2450 (23.3%) in controls and 820 (23.3%) in cases; and those with 0.515 to 11 were 2551 (24.2%) in controls and 854 (24.2%) in cases.

### 2.2. Association of SNPs in Genes of the Citric Acid Cycle with the Risk of Colorectal Cancer

Table 3 shows the association between the citric acid cycle SNPs and colorectal cancer by subsites. Rs35494829 C > T in the *SUCLG2* gene exhibited a significant association with colorectal cancer (OR [95% CIs]; CT, 0.89 [0.80–0.98] compared to CC; CT + TT, 0.88 [0.80–0.97] compared to CC; per T allele, 0.88 [0.81–0.96]) and colon cancer (CT, 0.83 [0.73–0.94] compared to CC; CT + TT, 0.82 [0.72–0.92] compared to CC; per T allele, 0.88 [0.81–0.96]). rs7511595 T > C in the *OGDHL* gene had a significant association with a decreased risk of rectal cancer (CC, 0.60 [0.40–0.92] compared to TT). Rs10891314 G > A exhibited a significant association with colorectal cancer (AA, 0.82 [0.73–0.94] compared to GG; GA + AA, 0.92 [0.85–0.99] compared to GG; per A allele, 0.92 [0.87–0.97]) and colon cancer (AA, 0.75 [0.65–0.88] compared to GG; GA + AA, 0.88 [0.80–0.97] compared to GG; per A allele, 0.88 [0.82–0.95]).

### 2.3. Interaction between SNPs in Genes of the Citric Acid Cycle and Contributors to Energy Balance on the Risk of Colorectal Cancer

The interaction between SNPs in the gene encoding components of the citric acid cycle and contributors of energy balance on colorectal cancer risk was investigated. Odds ratios and corresponding 95% confidence intervals for environmental factors are presented by risk allele noncarrier and carrier only if the *p*-values of those interactions were under 0.05. Table 4 presents the results of the SNP-SNP interaction for colon cancer, showing that the 95% CIs of AP did not contain zero. Table 4 shows the OR and 95% CIs for the contributors of energy balance on the risk of colon cancer by noncarriers or carriers of mitochondrial SNPs showing an interaction *p*-Value under 0.05. Significant interactions comprising *SDHC*-rs17395595 and obesity (p for interaction = 0.0023), *MDH1*-rs2278718 and severe obesity (0.0229), *SUCLG2*-rs902320 and severe obesity (0.0437), *SUCLG2*-rs902321 and severe obesity (0.0071), *PCK2*-rs55733026 and energy intake (0.0376), and *ACLY*-rs2304497 and vigorous physical activity (0.0450) on colon cancer were observed. Obesity and severe obesity were significantly associated with colon cancer among noncarriers of *SDHC*-rs17395595 (1.42 [1.24–1.63]) and carriers of *SUCLG2*-rs902321 (1.74 [1.07–2.82]), respectively.

Odds ratios and corresponding 95% confidence intervals for energy balance-related environmental factors for rectal cancer by risk allele noncarrier and carrier are shown in Table 5. Significant interactions comprising *MDH1*-rs2278718 and obesity (p for interaction = 0.0450), *SUCLG2*-rs902321 and severe obesity (0.0468), *SUCLG2*-rs35494829 and severe obesity (0.0457), *SUCLG2*-rs35494829 and abdominal obesity (0.0159), *OGDHL*-rs11101224 and abdominal obesity (0.0193), and *OGDHL*-rs751595 and abdominal obesity (0.0056) on rectal cancer were observed. Abdominal obesity was significantly associated with rectal cancer among noncarriers of *SUCLG2*-rs35494829 (1.35 [1.12–1.63]), *OGDHL*-rs11101224 (1.37 [1.12–1.68]), and *OGDHL*-rs751595 (1.40 [1.14–1.72]). None of these association reached statistical significance after Bonferroni correction.

### 2.4. Pairwise SNP-SNP Interactions of SNPs within the Citric Acid Cycle on the Risk of Colorectal Cancer

Table 6 presents the results of the SNP-SNP interaction for colon cancer, showing that the 95% CIs of AP do not contain zero. The APs of the interaction comprising *SDHC*-rs17395595 and *IDH3A*-rs11555541 (−0.348 [−0.628–0.068]); *MDH1*-rs2278718 and *SUCLG2*-rs902321 (−0.301 [−0.525–0.077]); *IDH1*-rs34218846 and *IDH3A*-rs11555541 (−0.507 [−0.978–0.036]); *SUCLG2*-rs902320 and *IDH3A*-rs17674205 (−0.570 [−0.966–0.174]); *SUCLG2*-rs902321 and *OGDHL*-rs751595 (−0.259 [−0.503–0.015]); *SUCLG2*-rs902321 and *IDH3A*-rs11555541 (−0.258 [−0.500–0.016]); *SUCLG2*-rs902321 and *IDH3A*-rs17674205 (−0.491 [−0.862–0.121]); *SUCLG2*-rs35494829 and *IDH3A*-rs17674205 (−0.358 [−0.716–0.001]); *SUCLG2*-rs2363712 and *IDH3A*-rs11555541 (−0.282 [−0.508–0.055]); *SUCLG2*-rs2363712 and *IDH3A*-rs17674205 (−0.496 [−0.866–0.126]); *OGDHL*-rs11101224 and *OGDHL*-rs751595 (−0.440 [−0.857–0.022]); and *DLAT*-rs10891314 and *IDH3A*-rs11555541 (−0.288 [−0.530–0.046]) were found to have negative values and therefore could be less than additivity.

Table 7 shows the results of the SNP-SNP interaction for rectal cancer, showing that the 95% CIs of AP did not contain zero. APs between *SDHC*-rs17395595 and *IDH3A*-rs11555541 (−0.341 [−0.672–0.010]); *SUCLG2*-rs902320 and *SDHA*-rs6962 (−0.390 [−0.774–0.006]); *SUCLG2*-rs902321 and *ACO1*-rs7042042 (−0.431 [−0.784–0.078]); *SUCLG2*-rs2363712 and *ACO1*-rs7042042 (−0.368 [−0.681–0.054]); and *SDHA*-rs34511054 and *ACLY*-rs2304497 (−0.704 [−1.362–0.047]) were found to be negative, indicating that the interactions are less than additivity. None of these association reached statistical significance after Bonferroni correction.

## 3. Discussion

In this study, we evaluated the associations between polymorphisms in the citric acid cycle and colorectal cancer in UK populations. The interaction between the citric acid cycle marker and the contributors of energy balance, including obesity, physical activity, and energy intake, on the risk of colorectal cancer was examined. We found significant interactions between citric acid cycle SNPs and the risk of colon and rectal cancer.

We found the significant association between *SUCLG2* rs35494829 and colon cancer (ORs [95% CIs] per increment of the minor allele, 0.82 [0.74–0.92]). These results can be supported by few studies on succinate, which were catalyzed by succinyl-CoA ligase, as an intermediate in cancer metabolism. Results from an in vitro study have suggested that the accumulation of succinate leads to the oncogenic signal via HIF-1α regulatory pathway [12].

The citric acid cycle is the core metabolic pathway in mitochondria. Altered metabolite profiles of the citric acid cycle have been reported from recent metabolomic studies using serum [13] and urinary [14] samples from colorectal cancer patients. The underlying knowledge of the association between the genes encoding the enzymes of the citric acid cycle and cancer has usually been described as the Warburg effect [15], referring to the phenomenon that occurs in most cancer cells where energy is generated by lactic acid fermentation in the cytosol instead of pyruvate from glycolysis even when oxygen is sufficient [16,17]. Recent article has focused on targeting the citric acid cycle as the potential therapeutic strategy for cancer [18].

The *SDHC* gene encodes one of four nuclear-encoded subunits comprising the succinate dehydrogenase (SDH) enzyme, which links the citric acid cycle to oxidative phosphorylation within mitochondria. Dysfunction of the electron transport chain due to defects in SDH subunits B, C, and D has been found in patients with gastrointestinal stromal tumors [19]. The results from previous studies also reported that the expression of *SDHC* was reduced in tumor tissues [20,21,22]. Alteration of the *SDHC* gene leads to reduced SDH activity, increases the levels of mitochondrial succinate and then increases mitochondrial reactive oxygen species [22]. Recent studies have suggested that the consequences of dysfunctions in the genes encoding components of the SDH enzyme and fumarate hydratase (FH) were linked to mitochondrial dysfunction and cancers [23], with dysfunctional cell signaling via oncometabolites including succinate and fumarate [24], and there were similarities between the phenotypes of cancers with these mutations [25]. However, unlike expectations, the interaction between SNPs in the *SDHC* and FH genes in colorectal cancer was not found in the present study.

A few studies have reported associations between SNPs in the citric acid pathway and colorectal cancer [26,27]. Studies on the association of SNPs in the *SDHC* gene were conducted for the prognosis of patients with colorectal cancer [26]. The significant associations were rs12064957 (1.36 [1.06–1.74]) for overall survival in the additive model and rs413826 for overall survival and recurrence-free survival (0.61 [0.47–0.79] and 0.73 [0.58–0.91], respectively) in the additive model [26]. The *ACLY* gene has been evaluated for the prognosis and survival of colorectal cancer [27] in the Chinese population. Rs2304497 and rs9912300 in the *ACLY* gene showed significant associations with the risks of death (HR [95% CIs]; 0.47 [0.24–0.90] and 0.59 [0.37–0.92], respectively) and recurrence (0.46 [0.24–0.86] and 0.54 [0.35–0.83], respectively) in patients with stage III + IV colorectal cancer [27]. Although the *SDHD* gene was not included in this study, it has previously been reported for its association with colorectal cancer [26]. Rs544184 and rs7121782 showed a significant association with overall survival (HR [95% CIs]; 1.52 [1.05–2.19] and 1.49 [1.04–2.14] in the additive model, respectively). Rs10789859, rs544184, and rs7121782 exhibited a significant association with recurrence-free survival (HR [95% CIs]; 1.29 [1.08–1.55], 1.31 [1.08–1.58] and 1.29 [1.07–1.55] in the additive model, respectively).

Although we anticipated to provide clues to the etiology in cancer development related to energy metabolism through the results of the present study based on a large population, the causality still remains inconclusive. Thus, further studies were required to ensure the associations which were exhibited in this study and identify more precise mechanisms considering the potential confounders. A prospective study defined the subtype of colorectal cancer based on microsatellite instability, CpG island methylator, and BRAF mutation, which is also referred to as proto-oncogene B-Raf, and found that the association between smoking and colorectal cancer differs according to the molecular subtype of colorectal cancer among women [28].

## 4. Materials and Methods

### 4.1. Study Population

The sample consisted of people who participated in the UK Biobank study. The UK Biobank was initially set up as a national resource to study lifestyle and genetic factors affecting aging traits at a population level. Participants were registered with the UK National Health Service and lived within 25 miles of one of the 22 assessment centers. More than 500,000 volunteers attended across the UK from 2007 to 2013. Participants donated samples for genotyping and completed lifestyle questionnaires under standard measurements. The UK Biobank resource is described extensively elsewhere [29,30].

A total of 483,149 participants remained after the exclusion of participants with no information on cancer incidence, genotype, ethnic background, or socioeconomic deprivation, and 3637 participants were identified with incident cases of colorectal cancer (Figure 1). For each case, three controls were selected using incidence density sampling [31] from participants who had been diagnosed with colorectal cancer with matching on sex, age group at enrollment by five years, ethnic background (British, Irish or any other white background classified as White, mixed ethnic background between White and Black Caribbean, White and Black African, White and Asian or any other mixed background classified as mixed; Indian, Pakistani, Bangladeshi or any other Asian background classified as Asian or Asian British; Caribbean, African or any other Black background classified as Black or Black British; Chinese; other), the 22 study centers (Barts, Birmingham, Bristol, Bury, Cardiff, Croydon, Edinburgh, Glasgow, Hounslow, Leeds, Liverpool, Manchester, Middlesbrough, Newcastle, Nottingham, Oxford, Reading, Sheffield, Stockport, Stoke, Swansea, Wrexham) and the Townsend deprivation index at recruitment (divided into four groups by quartiles).

### 4.2. Data Collection and Measurements

Body size, including waist and hip circumference, height, and weight, was directly measured at enrollment. Obesity was defined using body mass index (BMI) and the waist-to-hip ratio (WHR). The participants were classified as people with obesity who had greater than or equal to 30 kg/m^2^ BMI and as people with severe obesity who had greater than or equal to 40 kg/m^2^ BMI. The waist-to-hip ratio was also used to assess obesity [32]. Men with a WHR over 0.9 and women with a WHR over 0.85 were classified as participants with abdominal obesity [32].

Energy intake was estimated as nutrient intake via the diet by 24-h recall with the units of kJ. An estimated amount of daily energy consumption of more than 2000 kilocalories a day for women and 2500 for men was classified as excess energy intake. Participants also reported information on physical activities, including the number of days per week of moderate/vigorous physical activity for more than 10 min and the duration of moderate/vigorous activity on a typical day. Participants who performed over 150 min of moderate physical activity or 75 min of vigorous physical activity throughout the week were classified as people who achieved physical activity for general health benefits.

Townsend deprivation index scores were used from national census data about car ownership, household overcrowding, owner-occupation, and unemployment aggregated for postcodes of residence [33]. Higher Townsend scores were associated with higher levels of socioeconomic deprivation. The Townsend deprivation index categorized by quartile among both controls and cases was included in the analyses.

### 4.3. Outcome Ascertainment

The UK Biobank obtained data on cancer diagnoses and deaths through the National Health Service (NHS) Digital for participants in England and Wales and the NHS Central Register for participants in Scotland. The completeness of case ascertainment in English cancer registries is reported to be approximately 98–99%, based on a study that linked routine cancer registration with information from the Hospital Episode Statistics database [34]. Colorectal, colon, and rectal cancers were classified according to the International Classification of Diseases (C18-C20, C18, and C19-C20, respectively) only for cancer diagnosed after enrollment.

### 4.4. Genotyping

Participants answered detailed questionnaires on lifestyles, had measurements taken, and provided blood, urine, and saliva samples. Two arrays with over 95% common marker content were used for genotyping the individuals. The UK Biobank data release available at the time of analysis included genotypes for 488,377 participants obtained through either the custom UK Biobank Axiom array or the Affymetrix Axiom Array. Genotypes imputed to the Haplotype Reference Consortium 48 and the combined UK10K/1000 Genomes panels were retrieved from the UK Biobank data showcase [35].

### 4.5. Marker Selection

The MitoProteome database (available at http://www.mitoproteome.org/) was used to select the genes contributing to the citric acid cycle [36]. We find the citric acid cycle gene based on Kyoto encyclopedia of genes and genomes (KEGG) [37] using the keyword of “Citrate cycle (TCA cycle)” and 27 autosomal genes were extracted. Then, SNPs within the 27 genes related to the TCA cycle were found using the dbSNP database [38]. SNPs related to the citric acid cycle were selected based on the following criteria: (1) genetic variant on the mitochondrial citric acid cycle; (2) functionally important variant that might affect gene transcript structure or protein, such as coding nonsynonymous SNPs or SNPs at a splicing site; (3) common variant allele with minor allele frequency (MAF) > 5%; and (4) genotype call rate > 99%. Table 1 describes the information on SNPs that met the inclusion criteria.

### 4.6. Statistical Analyses

Conditional logistic regression was used to estimate the odds ratios (ORs) and corresponding 95% confidence intervals (CIs) of SNPs related to the citric acid cycle in additive and dominant models of colorectal cancer by subsites. The departure from Hardy–Weinberg equilibrium (HWE) among controls was assessed through Pearson’s chi-squared test. Genotypes of SNPs were dichotomized to noncarrier and carrier of the minor allele in the analyses of the gene-environment interactions and the SNP-SNP interactions. P for an interaction was calculated using the likelihood ratio test. Stratified analyses were also conducted by the number of minor alleles only when the interaction *p*-value was under 0.05. Pairwise SNP-SNP interactions were evaluated using the relative excess risk due to interaction (RERI) and the attributable proportion due to interaction (AP) [39]. SNPs were dichotomized, assuming the dominant model in the analyses on gene-environment and SNP-SNP interactions. RERI describes the effect due to interaction between two dichotomous risk factors, calculated with the following equation:(1)$$\mathrm{RERI}={\mathrm{RR}}_{E1+E2+}-{\mathrm{RR}}_{E1+E2-}-{\mathrm{RR}}_{E1-E2+}+1$$

AP is the measure quantifying the proportion of the combined effect due to interaction, calculated with the following equation:(2)$$\mathrm{AP}\;=\;\frac{\mathrm{RERI}}{{\mathrm{RR}}_{E1+E2+}}$$

The value of AP ranged from −1 to +1. An AP greater than zero means a positive interaction or more than additivity. An AP of less than zero means a negative interaction or less than additivity. It is recommended to use the risk factors rather than the preventive factors when calculating RERI and AP [40]; therefore, if the main effect of a SNP was preventive (that is, OR < 1), carriers of the minor allele were considered as the reference category in the analyses of the SNP-SNP interactions. Two-sided *p*-values less than 0.05 were regarded as statistically significant. Statistical analyses were conducted using R software version 3.6.3. Pairwise linkage disequilibrium was assessed using Haploview software version 4.2 [41].

## 5. Conclusions

The results from this study show that genetic variations of the enzyme within the citrate cycle are significantly associated with colorectal cancer susceptibility in UK populations. Interactions of SNPs in the citric acid cycle with the contributors to energy balance and SNP-SNP interactions are also exhibited in UK populations. This study identified SNPs of the citric acid cycle as a diagnostic marker of colorectal cancer and implicated them in understanding altered metabolism and mediating the citric acid cycle in colorectal cancer. We expected to provide the evidence on to identify the population at high risk of colorectal cancer via the results from the present study.

## Figures and Tables

**Figure 1 cancers-12-02939-f001:**
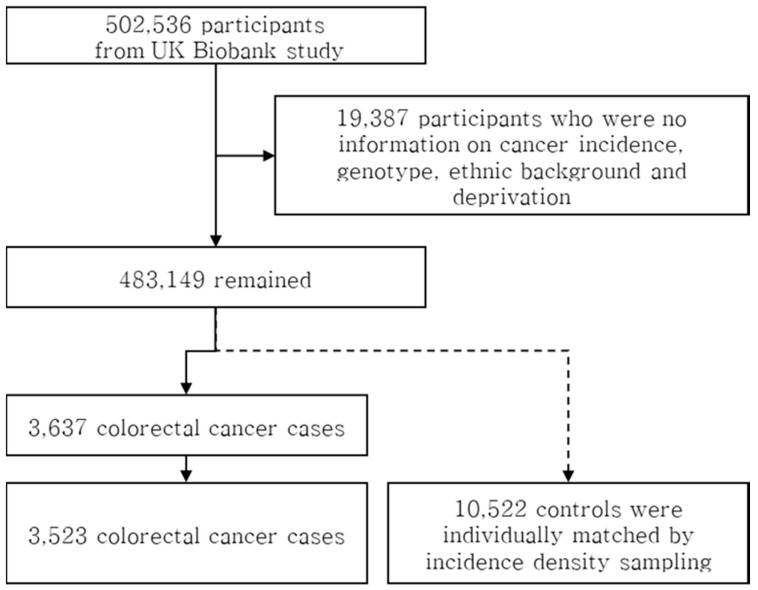
Flow chart of case and control selection.

**Table 1 cancers-12-02939-t001:** Information on SNPs that were included in the present study.

Gene	SNP	Chr: Position	Allele (a < A)	MAF		*p* for HWE	Call Rate (%)
				Control	CRC Case		
*SDHC*	rs16832884	1: 161368670	G < A	0.061	0.063	0.883	99.7
*SDHC*	rs17395595	1: 161374656	G < A	0.148	0.147	0.788	99.9
*MDH1*	rs2278718	2: 63588667	C < A	0.249	0.244	0.365	99.8
*IDH1*	rs34218846	2: 208243593	T < C	0.056	0.054	1.000	99.7
*SUCLG2*	rs902320	3: 67360679	T < C	0.270	0.261	0.451	99.9
*SUCLG2*	rs902321	3: 67360742	G < A	0.395	0.389	0.296	99.8
*SUCLG2*	rs35494829	3: 67375857	C < T	0.113	0.101	0.829	99.9
*SUCLG2*	rs2363712	3: 67376176	T < C	0.327	0.317	0.289	99.9
*SDHA*	rs6962	5: 256394	A < G	0.129	0.129	0.099	99.9
*SDHA*	rs34511054	5: 264041	C < A	0.059	0.061	0.651	99.8
*ACO1*	rs7042042	9: 32451146	A < G	0.356	0.356	0.740	99.9
*ACO1*	rs10970986	9: 32453280	C < T	0.291	0.294	0.919	99.9
*OGDHL*	rs11101224	10: 49742930	A < G	0.179	0.179	0.096	99.7
*OGDHL*	rs751595	10: 49756610	A < G	0.191	0.188	0.395	99.6
*DLAT*	rs10891314	11: 112045923	A < G	0.368	0.349	0.570	99.9
*PCK2*	rs55733026	14: 24095963	G < A	0.074	0.068	1.000	99.2
*PCK2*	rs1951634	14: 24100525	T < G	0.254	0.252	0.738	99.9
*PCK2*	rs35618680	14: 24103603	A < G	0.090	0.088	0.796	99.1
*IDH3A*	rs11555541	15: 78149427	C < T	0.495	0.495	0.418	99.9
*IDH3A*	rs17674205	15: 78169115	G < A	0.089	0.084	0.833	100.0
*ACLY*	rs8065502	17: 41892360	A < G	0.085	0.085	0.355	99.6
*ACLY*	rs2304497	17: 41909521	G < T	0.125	0.126	0.232	99.9

CRC, colorectal cancer; SNP, single nucleotide polymorphism; MAF, minor allele frequency, HWE, Hardy-Weinberg equilibrium. *p* values were calculated with Pearson’s χ2 tests; A and a were designated the major and minor alleles, respectively.

**Table 2 cancers-12-02939-t002:** The number of controls and patients with colorectal cancer in a nested case-control study from the participants in the UK Biobank.

Characteristics and Categories	Control, *n* (%)	Case, *n* (%)
N	10,522	3523
Age at enrollment		
36–40	24 (0.2)	8 (0.2)
41–45	272 (2.6)	93 (2.6)
46–50	575 (5.5)	193 (5.5)
51–55	1237 (11.8)	412 (11.7)
56–60	2007 (19.1)	673 (19.1)
61–65	3361 (31.9)	1123 (31.9)
66–70	3046 (28.9)	1021 (29.0)
Sex		
Men	6052 (57.5)	2024 (57.5)
Women	4470 (42.5)	1499 (42.5)
Ethnic background		
White	10,284 (97.7)	3423 (97.2)
Mixed	35 (0.3)	18 (0.5)
Asian or Asian British	83 (0.8)	31 (0.9)
Black or Black British	76 (0.7)	28 (0.8)
Chinese	11 (0.1)	6 (0.2)
Other	33 (0.3)	17 (0.5)
Assessment center at which participant consented		
Barts	230 (2.2)	76 (2.2)
Birmingham	399 (3.8)	135 (3.8)
Bristol	887 (8.4)	296 (8.4)
Bury	699 (6.6)	236 (6.7)
Cardiff	453 (4.3)	152 (4.3)
Croydon	408 (3.9)	136 (3.9)
Edinburgh	445 (4.2)	148 (4.2)
Glasgow	474 (4.5)	158 (4.5)
Hounslow	472 (4.5)	157 (4.5)
Leeds	928 (8.8)	309 (8.8)
Liverpool	673 (6.4)	226 (6.4)
Manchester	339 (3.2)	114 (3.2)
Middlesbrough	353 (3.4)	118 (3.3)
Newcastle	903 (8.6)	300 (8.5)
Nottingham	704 (6.7)	236 (6.7)
Oxford	378 (3.6)	128 (3.6)
Reading	682 (6.5)	229 (6.5)
Sheffield	572 (5.4)	193 (5.5)
Stockport	12 (0.1)	5 (0.1)
Stoke	460 (4.4)	154 (4.4)
Swansea	45 (0.4)	15 (0.4)
Wrexham	6 (0.1)	2 (0.1)
Townsend deprivation index at recruitment		
[−6.26, 3.65]	2788 (26.5)	933 (26.5)
(−3.65, 2.15]	2733 (26.0)	916 (26.0)
(−2.15, 0.515]	2450 (23.3)	820 (23.3)
(0.515, 11]	2551 (24.2)	854 (24.2)

**Table 3 cancers-12-02939-t003:** Odds ratios (OR) and 95% confidence intervals (CIs) of the association of citric acid cycle SNPs with the risk of colorectal cancer by subsites.

Gene-SNP	Colon Cancer			Rectal Cancer		
	Controls, *n* (%)	Cases, *n* (%)	OR (95% CIs)	Controls, *n* (%)	Cases, *n* (%)	OR (95% CIs)
*SDHC*-rs16832884						
CC	6183 (88.0)	2061 (87.6)	1.00 (reference)	3190 (88.7)	1059 (88.0)	1.00 (reference)
CT	816 (11.6)	285 (12.1)	1.04 (0.90–1.20)	393 (10.9)	140 (11.6)	1.08 (0.87–1.32)
TT	27 (0.4)	6 (0.3)	0.64 (0.26–1.55)	14 (0.4)	4 (0.3)	0.85 (0.28–2.59)
CT + TT			1.03 (0.89–1.19)			1.07 (0.87–1.31)
Per T allele			1.01 (0.88–1.16)			1.06 (0.87–1.29)
*SDHC*-rs17395595						
AA	5062 (72.4)	1680 (71.6)	1.00 (reference)	2616 (73.0)	895 (74.5)	1.00 (reference)
AG	1785 (25.5)	613 (26.1)	1.04 (0.94–1.16)	883 (24.6)	280 (23.3)	0.93 (0.80–1.08)
GG	142 (2.0)	52 (2.2)	1.11 (0.81–1.54)	84 (2.3)	26 (2.2)	0.89 (0.57–1.39)
AG + GG			1.05 (0.94–1.16)			0.93 (0.80–1.07)
Per G allele			1.05 (0.95–1.15)			0.93 (0.82–1.07)
*MDH1*-rs2278718						
GG	3991 (56.9)	1350 (57.4)	1.00 (reference)	2003 (55.7)	688 (57.3)	1.00 (reference)
GA	2580 (36.8)	857 (36.5)	0.98 (0.89–1.09)	1365 (38.0)	435 (36.2)	0.93 (0.81–1.07)
AA	444 (6.3)	143 (6.1)	0.95 (0.78–1.16)	228 (6.3)	78 (6.5)	1.00 (0.77–1.32)
GA + AA			0.98 (0.89–1.08)			0.94 (0.83–1.07)
Per A allele			0.98 (0.91–1.06)			0.97 (0.87–1.07)
*IDH1*-rs34218846						
GG	6252 (89.0)	2100 (89.3)	1.00 (reference)	3212 (89.4)	1080 (89.9)	1.00 (reference)
GA	750 (10.7)	245 (10.4)	0.98 (0.84–1.14)	368 (10.2)	117 (9.7)	0.95 (0.77–1.18)
AA	19 (0.3)	6 (0.3)	0.93 (0.37–2.33)	13 (0.4)	5 (0.4)	1.07 (0.38–3.03)
GA + AA			0.97 (0.84–1.13)			0.96 (0.77–1.18)
Per A allele			0.97 (0.84–1.13)			0.96 (0.79–1.18)
*SUCLG2*-rs902320						
GG	3753 (53.4)	1280 (54.5)	1.00 (reference)	1921 (53.5)	650 (54.0)	1.00 (reference)
GA	2730 (38.9)	907 (38.6)	0.97 (0.88–1.08)	1415 (39.4)	476 (39.6)	0.99 (0.87–1.14)
AA	541 (7.7)	162 (6.9)	0.87 (0.72–1.05)	256 (7.1)	77 (6.4)	0.89 (0.69–1.17)
GA + AA			0.96 (0.87–1.05)			0.98 (0.86–1.11)
Per A allele			0.95 (0.88–1.03)			0.97 (0.87–1.07)
*SUCLG2*-rs902321						
TT	2605 (37.2)	870 (37.1)	1.00 (reference)	1315 (36.6)	439 (36.7)	1.00 (reference)
TG	3282 (46.9)	1118 (47.7)	1.02 (0.92–1.13)	1701 (47.4)	586 (49.0)	1.03 (0.89–1.19)
GG	1114 (15.9)	358 (15.3)	0.96 (0.84–1.11)	572 (15.9)	172 (14.4)	0.90 (0.74–1.10)
TG + GG			1.00 (0.91–1.10)			1.00 (0.87–1.14)
Per G allele			0.99 (0.92–1.06)			0.97 (0.88–1.06)
*SUCLG2*-rs35494829						
CC	5516 (78.7)	1919 (81.9)	1.00 (reference)	2812 (78.7)	945 (78.8)	1.00 (reference)
CT	1402 (20.0)	404 (17.3)	0.83 (0.73–0.94)	715 (20.0)	241 (20.1)	1.00 (0.85–1.18)
TT	87 (1.2)	19 (0.8)	0.64 (0.39–1.05)	47 (1.3)	14 (1.2)	0.89 (0.49–1.63)
CT + TT			0.82 (0.72–0.92)			1.00 (0.85–1.17)
Per T allele			0.82 (0.74–0.92)			0.99 (0.86–1.15)
*SUCLG2*-rs2363712						
TT	3184 (45.4)	1087 (46.3)	1.00 (reference)	1643 (45.8)	560 (46.6)	1.00 (reference)
TC	3048 (43.5)	1018 (43.4)	0.98 (0.89–1.08)	1561 (43.5)	531 (44.1)	1.00 (0.87–1.14)
CC	777 (11.1)	242 (10.3)	0.91 (0.77–1.07)	386 (10.8)	112 (9.3)	0.86 (0.68–1.08)
TC + CC			0.97 (0.88–1.06)			0.97 (0.85–1.10)
Per C allele			0.96 (0.90–1.03)			0.95 (0.86–1.05)
*SDHA*-rs6962						
GG	5297 (75.5)	1787 (76.0)	1.00 (reference)	2726 (75.9)	896 (74.5)	1.00 (reference)
GT	1605 (22.9)	529 (22.5)	0.98 (0.87–1.09)	808 (22.5)	294 (24.4)	1.10 (0.94–1.28)
TT	110 (1.6)	34 (1.4)	0.92 (0.62–1.35)	57 (1.6)	13 (1.1)	0.68 (0.37–1.26)
GT + TT			0.97 (0.87–1.09)			1.08 (0.92–1.25)
Per T allele			0.97 (0.88–1.07)			1.04 (0.90–1.19)
*SDHA*-rs34511054						
GG	6202 (88.5)	2063 (88.0)	1.00 (reference)	3176 (88.6)	1065 (88.7)	1.00 (reference)
GA	776 (11.1)	275 (11.7)	1.07 (0.92–1.24)	399 (11.1)	130 (10.8)	0.97 (0.79–1.20)
AA	31 (0.4)	7 (0.3)	0.67 (0.30–1.53)	9 (0.3)	6 (0.5)	1.99 (0.71–5.60)
GA + AA			1.05 (0.91–1.22)			1.00 (0.81–1.22)
Per A allele			1.03 (0.90–1.19)			1.02 (0.84–1.24)
*ACO1*-rs7042042						
GG	2898 (41.3)	946 (40.2)	1.00 (reference)	1501 (41.8)	515 (42.8)	1.00 (reference)
GA	3241 (46.2)	1110 (47.2)	1.05 (0.95–1.16)	1619 (45.1)	544 (45.2)	0.98 (0.85–1.13)
AA	880 (12.5)	295 (12.5)	1.02 (0.88–1.19)	467 (13.0)	144 (12.0)	0.90 (0.73–1.11)
GA + AA			1.04 (0.95–1.15)			0.96 (0.84–1.10)
Per A allele			1.02 (0.95–1.10)			0.96 (0.87–1.05)
*ACO1*-rs10970986						
AA	3561 (50.8)	1174 (50.0)	1.00 (reference)	1776 (49.5)	586 (48.7)	1.00 (reference)
AG	2850 (40.6)	992 (42.2)	1.06 (0.96–1.17)	1510 (42.0)	501 (41.6)	1.01 (0.88–1.16)
GG	603 (8.6)	183 (7.8)	0.93 (0.78–1.11)	305 (8.5)	116 (9.6)	1.15 (0.91–1.45)
AG + GG			1.04 (0.94–1.14)			1.03 (0.91–1.18)
Per G allele			1.00 (0.93–1.08)			1.05 (0.95–1.16)
*OGDHL*-rs11101224						
AA	4690 (66.8)	1557 (66.2)	1.00 (reference)	2438 (67.9)	825 (68.5)	1.00 (reference)
AG	2121 (30.2)	717 (30.5)	1.02 (0.92–1.13)	1046 (29.1)	351 (29.2)	0.99 (0.86–1.15)
GG	209 (3.0)	77 (3.3)	1.12 (0.85–1.46)	109 (3.0)	28 (2.3)	0.76 (0.49–1.16)
AG + GG			1.03 (0.93–1.13)			0.97 (0.84–1.12)
Per G allele			1.03 (0.94–1.12)			0.95 (0.84–1.08)
*OGDHL*-rs751595						
TT	4589 (65.5)	1529 (65.2)	1.00 (reference)	2353 (65.5)	798 (66.3)	1.00 (reference)
TC	2174 (31.0)	734 (31.3)	1.01 (0.91–1.12)	1101 (30.7)	378 (31.4)	1.01 (0.88–1.17)
CC	243 (3.5)	83 (3.5)	1.01 (0.78–1.31)	136 (3.8)	28 (2.3)	0.60 (0.40–0.92)
TC + CC			1.01 (0.92–1.12)			0.97 (0.84–1.11)
Per C allele			1.01 (0.93–1.10)			0.93 (0.82–1.05)
*DLAT*-rs10891314						
GG	2759 (39.6)	996 (42.7)	1.00 (reference)	1442 (40.5)	486 (40.8)	1.00 (reference)
GA	3231 (46.4)	1073 (46.0)	0.92 (0.83–1.02)	1672 (46.9)	557 (46.8)	0.99 (0.86–1.14)
AA	975 (14.0)	266 (11.4)	0.75 (0.65–0.88)	450 (12.6)	148 (12.4)	0.98 (0.79–1.22)
GA + AA			0.88 (0.80–0.97)			0.99 (0.86–1.13)
Per A allele			0.88 (0.82–0.95)			0.99 (0.90–1.09)
*PCK2*-rs55733026						
AA	6050 (86.2)	2048 (87.1)	1.00 (reference)	3060 (85.1)	1042 (86.8)	1.00 (reference)
AG	937 (13.3)	296 (12.6)	0.93 (0.81–1.07)	516 (14.4)	147 (12.2)	0.83 (0.68–1.01)
GG	35 (0.5)	8 (0.3)	0.68 (0.32–1.47)	18 (0.5)	12 (1.0)	2.03 (0.95–4.33)
AG + GG			0.92 (0.80–1.06)			0.87 (0.72–1.05)
Per G allele			0.92 (0.80–1.05)			0.92 (0.77–1.10)
*PCK2*-rs1951634						
CC	3918 (55.9)	1314 (56.0)	1.00 (reference)	1993 (55.6)	669 (55.7)	1.00 (reference)
CT	2652 (37.8)	881 (37.5)	0.99 (0.89–1.09)	1351 (37.7)	460 (38.3)	1.02 (0.89–1.17)
TT	444 (6.3)	152 (6.5)	1.01 (0.84–1.23)	243 (6.8)	72 (6.0)	0.88 (0.67–1.17)
CT + TT			0.99 (0.90–1.09)			1.00 (0.87–1.14)
Per T allele			1.00 (0.92–1.08)			0.98 (0.88–1.09)
*PCK2*-rs35618680						
GG	5798 (82.7)	1953 (83.3)	1.00 (reference)	2976 (83.0)	998 (83.2)	1.00 (reference)
GA	1156 (16.5)	370 (15.8)	0.95 (0.84–1.08)	580 (16.2)	194 (16.2)	0.99 (0.83–1.19)
AA	56 (0.8)	21 (0.9)	1.10 (0.67–1.83)	31 (0.9)	8 (0.7)	0.76 (0.35–1.66)
GA + AA			0.96 (0.85–1.09)			0.98 (0.82–1.17)
Per A allele			0.97 (0.86–1.09)			0.97 (0.83–1.14)
*IDH3A*-rs11555541						
AA	1823 (26.0)	600 (25.5)	1.00 (reference)	903 (25.2)	309 (25.7)	1.00 (reference)
AC	3447 (49.1)	1178 (50.1)	1.04 (0.93–1.16)	1814 (50.5)	595 (49.4)	0.96 (0.82–1.12)
CC	1749 (24.9)	575 (24.4)	1.00 (0.87–1.14)	872 (24.3)	300 (24.9)	1.00 (0.84–1.21)
AC + CC			1.03 (0.92–1.14)			0.97 (0.84–1.13)
Per C allele			1.00 (0.94–1.07)			1.00 (0.91–1.10)
*IDH3A*-rs17674205						
TT	5788 (83.1)	1957 (84.0)	1.00 (reference)	2949 (82.7)	1008 (84.1)	1.00 (reference)
TC	1127 (16.2)	357 (15.3)	0.94 (0.83–1.07)	588 (16.5)	180 (15.0)	0.90 (0.75–1.08)
CC	51 (0.7)	16 (0.7)	0.96 (0.54–1.70)	28 (0.8)	10 (0.8)	1.09 (0.53–2.26)
TC + CC			0.94 (0.83–1.07)			0.91 (0.76–1.08)
Per C allele			0.95 (0.84–1.07)			0.92 (0.78–1.09)
*ACLY*-rs8065502						
AA	5890 (83.9)	1959 (83.3)	1.00 (reference)	3009 (83.8)	1017 (84.5)	1.00 (reference)
AG	1060 (15.1)	377 (16.0)	1.07 (0.94–1.21)	561 (15.6)	180 (15.0)	0.95 (0.79–1.14)
GG	67 (1.0)	16 (0.7)	0.72 (0.42–1.25)	21 (0.6)	7 (0.6)	0.99 (0.42–2.32)
AG + GG			1.04 (0.92–1.18)			0.95 (0.79–1.14)
Per G allele			1.02 (0.91–1.15)			0.95 (0.80–1.13)
*ACLY*-rs2304497						
GG	5414 (77.2)	1790 (76.2)	1.00 (reference)	2721 (75.7)	923 (76.7)	1.00 (reference)
GA	1482 (21.1)	524 (22.3)	1.07 (0.95–1.20)	813 (22.6)	258 (21.4)	0.94 (0.80–1.10)
AA	121 (1.7)	35 (1.5)	0.88 (0.60–1.28)	60 (1.7)	22 (1.8)	1.09 (0.66–1.78)
GA + AA			1.05 (0.94–1.18)			0.95 (0.82–1.11)
Per A allele			1.03 (0.93–1.14)			0.97 (0.84–1.11)

**Table 4 cancers-12-02939-t004:** Odds ratios (OR) and 95% confidence intervals (CIs) for the contributors of energy balance on the risk of colon cancer by noncarriers or carriers of the citric acid cycle SNPs showing an interaction *p*-value under 0.05.

Environmental Variable	Noncarriers			Carriers			*P* _interaction_
	Controls, *n* (%)	Cases, *n* (%)	OR (95% CIs)	Controls, *n* (%)	Cases, *n* (%)	OR (95% CIs)	
	*SDHC*-rs17395595, G < A						
Obesity, BMI							0.0023
<30 kg/m^2^	3833 (75.9)	1163 (69.8)	1.00 (reference)	1416 (73.7)	495 (74.5)	1.00 (reference)	
≥30 kg/m^2^	1216 (24.1)	504 (30.2)	1.42 (1.24–1.63)	506 (26.3)	169 (25.5)	0.92 (0.69–1.23)	
	*MDH1*-rs2278718, C < A						
Severe obesity, BMI							0.0229
<40 kg/m^2^	3906 (98.1)	1317 (98.4)	1.00 (reference)	2973 (98.6)	973 (97.5)	1.00 (reference)	
≥40 kg/m^2^	76 (1.9)	21 (1.6)	1.18 (0.67–2.07)	42 (1.4)	25 (2.5)	1.47 (0.76–2.84)	
	*SUCLG2*-rs902320, T < C						
Severe obesity, BMI							0.0437
<40 kg/m^2^	3672 (98.0)	1250 (98.3)	1.00 (reference)	3216 (98.7)	1039 (97.7)	1.00 (reference)	
≥40 kg/m^2^	76 (2.0)	22 (1.7)	0.89 (0.50–1.59)	42 (1.3)	24 (2.3)	1.95 (1.00–3.81)	
	*SUCLG2*-rs902321, G < A						
Severe obesity, BMI							0.0071
<40 kg/m^2^	2544 (97.8)	851 (98.6)	1.00 (reference)	4321 (98.6)	1435 (97.7)	1.00 (reference)	
≥40 kg/m^2^	58 (2.2)	12 (1.4)	0.82 (0.39–1.69)	60 (1.4)	34 (2.3)	1.74 (1.07–2.82)	
	*PCK2*-rs55733026, G < A						
Daily energy intake, kcal							0.0376
Men, ≤ 0.9; women, ≤ 0.85	340 (47.0)	99 (43.4)	1.00 (reference)	61 (50.8)	21 (52.5)	1.00 (reference)	
Men, > 0.9; women, > 0.85	383 (53.0)	129 (56.6)	1.22 (0.79–1.90)	59 (49.2)	19 (47.5)	0.86 (0.45–1.45)	
	*ACLY*-rs2304497, G < T						
Vigorous physical activity							0.045
Not sufficient	1533 (54.0)	428 (49.7)	1.00 (reference)	422 (50.0)	134 (51.3)	1.00 (reference)	
Sufficient	1304 (46.0)	434 (50.3)	1.19 (0.98–1.46)	422 (50.0)	127 (48.7)	1.44 (0.75–2.78)	

BMI, body mass index; WHR, waist-hip ratio. Statistical evaluation for interaction was made based on the likelihood ratio test with one degree of freedom. An interaction term was created between dichotomized variables representing genotypes and environmental factors.

**Table 5 cancers-12-02939-t005:** Odds ratios (OR) and 95% confidence intervals (CIs) for the contributors of energy balance on the risk of rectal cancer by noncarriers and carriers of the citric acid cycle SNPs showing an interaction *p*-value under 0.05.

Environmental Variable	Noncarriers			Carriers			*P* _interaction_
	Controls, *n* (%)	Cases, *n* (%)	OR (95% CIs)	Controls, *n* (%)	Cases, *n* (%)	OR (95% CIs)	
	*MDH1*-rs2278718, C < A						
Obesity, BMI							0.045
<30 kg/m^2^	1503 (75.3)	528 (76.7)	1.00 (reference)	1227 (77.1)	370 (72.8)	1.00 (reference)	
≥30 kg/m^2^	493 (24.7)	160 (23.3)	0.91 (0.72–1.15)	364 (22.9)	138 (27.2)	1.39 (1.04–1.87)	
	*SUCLG2*-rs902321, G < A						
Severe obesity, BMI							0.0468
<40 kg/m^2^	1283 (97.7)	432 (98.6)	1.00 (reference)	2240 (98.9)	739 (98.0)	1.00 (reference)	
≥40 kg/m^2^	30 (2.3)	6 (1.4)	0.40 (0.13–1.24)	26 (1.1)	15 (2.0)	1.38 (0.67–2.84)	
	*SUCLG2*-rs35494829, C < T						
Severe obesity, BMI							0.0457
<40 kg/m^2^	2766 (98.6)	921 (98.0)	1.00 (reference)	742 (97.8)	253 (99.2)	1.00 (reference)	
≥40 kg/m^2^	40 (1.4)	19 (2.0)	1.51 (0.83–2.75)	17 (2.2)	2 (0.8)	0.39 (0.04–3.78)	
Abdominal obesity, WHR							0.0159
Men, ≤0.9; women, ≤0.85	1152 (41.0)	333 (35.4)	1.00 (reference)	290 (38.1)	107 (42.0)	1.00 (reference)	
Men, >0.9; women, >0.85	1658 (59.0)	609 (64.6)	1.35 (1.12–1.63)	471 (61.9)	148 (58.0)	1.11 (0.64–1.94)	
	*OGDHL*-rs11101224, A < G						
Abdominal obesity, WHR							0.0193
Men, ≤0.9; women, ≤0.85	1003 (41.2)	288 (35.0)	1.00 (reference)	448 (38.8)	154 (40.7)	1.00 (reference)	
Men, >0.9; women, >0.85	1432 (58.8)	535 (65.0)	1.37 (1.12–1.68)	707 (61.2)	224 (59.3)	1.08 (0.75–1.55)	
	*OGDHL*-rs751595, G < A						
Abdominal obesity, WHR							0.0056
Men, ≤0.9; women, ≤0.85	977 (41.6)	276 (34.7)	1.00 (reference)	474 (38.3)	166 (41.0)	1.00 (reference)	
Men, >0.9; women, >0.85	1374 (58.4)	520 (65.3)	1.40 (1.14–1.72)	762 (61.7)	239 (59.0)	0.96 (0.68–1.35)	

BMI, body mass index; WHR, waist-hip ratio. Statistical evaluation for interaction was made based on the likelihood ratio test with one degree of freedom. An interaction term was created between dichotomized variables representing genotypes and environmental factors

**Table 6 cancers-12-02939-t006:** Odds ratios (OR) and 95% confidence intervals (CIs) for the combined effect of the citric acid cycle SNPs on the risk of colon cancer, showing the 95% CIs of AP not containing zero.

Gene-SNP	Gene-SNP	Controls, *n* (%)	Cases, *n* (%)	ORs (95% CIs)	AP (95% CIs)
*SDHC*-rs17395595	*IDH3A*-rs11555541				−0.348 (−0.628–0.068)
AA	AA	1357 (19.4)	414 (17.7)	1.00 (reference)	
AG + GG	AA	455 (6.5)	184 (7.8)	1.33 (1.09–1.63)	
AA	AC + CC	3698 (53.0)	1266 (54.0)	1.12 (0.99–1.27)	
AG + GG	AC + CC	1471 (21.1)	481 (20.5)	1.08 (0.93–1.25)	
*MDH1*-rs2278718	*SUCLG2*-rs902321				−0.301 (−0.525–0.077)
GG	TT	1484 (21.2)	463 (19.8)	1.00 (reference)	
GA + AA	TT	1117 (16.0)	407 (17.4)	1.17 (1.00–1.37)	
GG	TG + GG	2492 (35.7)	881 (37.6)	1.13 (1.00–1.29)	
GA + AA	TG + GG	1896 (27.1)	592 (25.3)	1.00 (0.87–1.15)	
*IDH1*-rs34218846	*IDH3A*-rs11555541				−0.507 (−0.978–0.036)
GG	AA	1638 (23.4)	522 (22.2)	1.00 (reference)	
GA + AA	AA	183 (2.6)	78 (3.3)	1.33 (1.00–1.76)	
GG	AC + CC	4606 (65.7)	1578 (67.1)	1.07 (0.96–1.20)	
GA + AA	AC + CC	586 (8.4)	173 (7.4)	0.93 (0.76–1.13)	
*SUCLG2*-rs902320	*IDH3A*-rs17674205				−0.570 (−0.966–0.174)
GG	TT	3140 (45.1)	1044 (44.9)	1.00 (reference)	
GA + AA	TT	2645 (38.0)	909 (39.1)	1.03 (0.93–1.15)	
GG	TC + CC	579 (8.3)	222 (9.5)	1.17 (0.98–1.39)	
GA + AA	TC + CC	599 (8.6)	151 (6.5)	0.76 (0.63–0.93)	
*SUCLG2*-rs902321	*OGDHL*-rs751595				−0.259 (−0.503–0.015)
TG + GG	TT	2920 (41.8)	948 (40.5)	1.00 (reference)	
TT	TT	1656 (23.7)	578 (24.7)	1.08 (0.96–1.22)	
TG + GG	TC + CC	1462 (20.9)	525 (22.4)	1.10 (0.97–1.25)	
TT	TC + CC	942 (13.5)	288 (12.3)	0.94 (0.81–1.09)	
*SUCLG2*-rs902321	*IDH3A*-rs11555541				−0.258 (−0.500–0.016)
TG + GG	AA	1169 (16.7)	359 (15.3)	1.00 (reference)	
TT	AA	648 (9.3)	240 (10.2)	1.20 (0.99–1.44)	
TG + GG	AC + CC	3222 (46.1)	1117 (47.6)	1.12 (0.98–1.29)	
TT	AC + CC	1954 (27.9)	630 (26.9)	1.05 (0.91–1.22)	
*SUCLG2*-rs902321	*IDH3A*-rs17674205				−0.491 (−0.862–0.121)
TT	TT	2174 (31.3)	703 (30.3)	1.00 (reference)	
TG + GG	TT	3590 (51.7)	1250 (53.8)	1.07 (0.96–1.19)	
TT	TC + CC	407 (5.9)	158 (6.8)	1.21 (0.98–1.48)	
TG + GG	TC + CC	769 (11.1)	212 (9.1)	0.86 (0.72–1.02)	
*SUCLG2*-rs35494829	*IDH3A*-rs17674205				−0.358 (−0.716–0.001)
CT + TT	TT	1254 (18.1)	342 (14.7)	1.00 (reference)	
CC	TT	4515 (65.0)	1606 (69.2)	1.31 (1.14–1.49)	
CT + TT	TC + CC	223 (3.2)	76 (3.3)	1.26 (0.94–1.68)	
CC	TC + CC	952 (13.7)	296 (12.8)	1.15 (0.97–1.38)	
*SUCLG2*-rs2363712	*IDH3A*-rs11555541				−0.282 (−0.508–0.055)
TC + CC	AA	1038 (14.8)	305 (13.0)	1.00 (reference)	
TT	AA	780 (11.1)	295 (12.6)	1.27 (1.06–1.53)	
TC + CC	AC + CC	2784 (39.8)	955 (40.7)	1.16 (1.00–1.35)	
TT	AC + CC	2399 (34.3)	792 (33.7)	1.12 (0.96–1.30)	
*SUCLG2*-rs2363712	*IDH3A*-rs17674205				−0.496 (−0.866–0.126)
TT	TT	2651 (38.1)	878 (37.8)	1.00 (reference)	
TC + CC	TT	3126 (45.0)	1074 (46.2)	1.04 (0.94–1.15)	
TT	TC + CC	507 (7.3)	196 (8.4)	1.17 (0.98–1.41)	
TC + CC	TC + CC	666 (9.6)	176 (7.6)	0.81 (0.68–0.97)	
*OGDHL*-rs11101224	*OGDHL*-rs751595				−0.440 (−0.857–0.022)
AG + GG	TT	294 (4.2)	82 (3.5)	1.00 (reference)	
AA	TT	4292 (61.3)	1446 (61.7)	1.20 (0.93–1.55)	
AG + GG	TC + CC	2031 (29.0)	709 (30.2)	1.24 (0.96–1.61)	
AA	TC + CC	382 (5.5)	107 (4.6)	1.00 (0.72–1.39)	
*DLAT*-rs10891314	*IDH3A*-rs11555541				−0.288 (−0.530–0.046)
GG	AA	703 (10.1)	223 (9.6)	1.00 (reference)	
GA + AA	AA	1104 (15.9)	374 (16.0)	1.07 (0.88–1.30)	
GG	AC + CC	2051 (29.5)	773 (33.1)	1.19 (1.01–1.42)	
GA + AA	AC + CC	3099 (44.5)	965 (41.3)	0.98 (0.83–1.16)	

AP, attributable proportion due to interaction.

**Table 7 cancers-12-02939-t007:** Odds ratios (OR) and 95% confidence intervals (CIs) for the combined effect of the citric acid cycle SNPs on the risk of colon cancer, showing the 95% CIs of AP not containing zero.

Gene-SNP	Gene-SNP	Controls, *n* (%)	Cases, *n* (%)	ORs (95% CIs)	AP (95% CIs)
*SDHC*-rs17395595	*IDH3A*-rs11555541				−0.341 (−0.672–0.010)
AG + GG	AA	258 (7.2)	68 (5.7)	1.00 (reference)	
AA	AA	644 (18.0)	239 (19.9)	1.42 (1.04–1.93)	
AG + GG	AC + CC	707 (19.8)	238 (19.8)	1.29 (0.95–1.75)	
AA	AC + CC	1966 (55.0)	656 (54.6)	1.28 (0.96–1.69)	
*SUCLG2*-rs902320	*SDHA*-rs6962				−0.390 (−0.774–0.006)
GA + AA	GG	1273 (35.5)	395 (32.9)	1.00 (reference)	
GG	GG	1449 (40.4)	500 (41.6)	1.12 (0.96–1.30)	
GA + AA	GT + TT	392 (10.9)	158 (13.1)	1.30 (1.04–1.62)	
GG	GT + TT	472 (13.2)	149 (12.4)	1.02 (0.82–1.26)	
*SUCLG2*-rs902321	*ACO1*-rs7042042				−0.431 (−0.784–0.078)
TG + GG	GG	957 (26.7)	301 (25.2)	1.00 (reference)	
TT	GG	541 (15.1)	211 (17.6)	1.25 (1.01–1.53)	
TG + GG	GA + AA	1312 (36.7)	456 (38.1)	1.11 (0.94–1.32)	
TT	GA + AA	768 (21.5)	228 (19.1)	0.95 (0.78–1.16)	
*SUCLG2*-rs2363712	*ACO1*-rs7042042				−0.368 (−0.681–0.054)
TC + CC	GG	836 (23.4)	258 (21.5)	1.00 (reference)	
TT	GG	663 (18.5)	256 (21.3)	1.25 (1.02–1.53)	
TC + CC	GA + AA	1106 (30.9)	385 (32.0)	1.13 (0.94–1.35)	
TT	GA + AA	975 (27.2)	303 (25.2)	1.01 (0.83–1.22)	
*SDHA*-rs34511054	*ACLY*-rs2304497				−0.704 (−1.362–0.047)
GA + AA	GG	314 (8.8)	93 (7.8)	1.00 (reference)	
GG	GG	2398 (66.9)	830 (69.2)	1.17 (0.92–1.49)	
GA + AA	GA + AA	93 (2.6)	43 (3.6)	1.56 (1.02–2.40)	
GG	GA + AA	777 (21.7)	234 (19.5)	1.01 (0.77–1.33)	
*SDHC*-rs17395595	*IDH3A*-rs11555541				−0.348 (−0.628–0.068)
AA	AA	1357 (19.4)	414 (17.7)	1.00 (reference)	
AG + GG	AA	455 (6.5)	184 (7.8)	1.33 (1.09–1.63)	
AA	AC + CC	3698 (53.0)	1266 (54.0)	1.12 (0.99–1.27)	
AG + GG	AC + CC	1471 (21.1)	481 (20.5)	1.08 (0.93–1.25)	

AP, attributable proportion due to interaction.

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
