# Peer review of "Effect of Citric Acid Cycle Genetic Variants and Their Interactions with Obesity, Physical Activity and Energy Intake on the Risk of Colorectal Cancer: Results from a Nested Case-Control Study in the UK Biobank"

_cancers, 2020, doi:10.3390/cancers12102939_

Round 1

Reviewer 1 Report

Please include if feasible the performance status of the patients.

Please add if possible data regarding smoke habit (pack-year)since there is a relationship between smoking and colon cancer 

I suggest to include a reference about BMI

Please include a perspective and possible application of these findings

I suggest to include the following references for a more in depth discussion

Protein cell 2018;9(2):216-37 about the role of tricarboxylic acid in cancer

J Natl Cancer Inst 2010;102(14):1012-22 regarding the association of smoking with colon cancer

Since it concerns metabolic cancer pathways, I suggest to include on line 61-62 page 2 the following about the activation of HIF and its function:

Oncotarget 2019;10(66):7071-79

Curr Molec Med 2018;18(6):343-51

Author Response

Response to Reviewer 1 Comments

Point 1: Please include if feasible the performance status of the patients.

Response 1: As the reviewer commented, we agree that the quality of our study results could be improved if the data have included the information on the performance status of the patients. However, currently available the performance status of the patients was not included in the provided data, and thus could not be attached.

Point 2. Please add if possible data regarding smoke habit (pack-year)since there is a relationship between smoking and colon cancer

Response 2: As the reviewer mentioned, we also agree that smoking is well-known risk factor for colorectal cancer. To reflect the comment of the reviewers, information on the frequency of smoking in subjects has been added to the Dicussion Section on page 3, lines 91-94, as described below.

“In addition to the matching variable, the proportion of never smoker was higher among cases than among controls (number and proportion of never, previous and current smoker, respectively; 5,315 [50.5], 4,162 [39.6] and 1,002 [9.5] among controls;1,607 [45.6], 1,550 [44.0] and 350 [9.9] among cases).”

Point 3. I suggest to include a reference about BMI

Response 3. To reflect the review’s suggestion, we have added the reference to the first paragraph in “Data collection and measurement”.

“WHO. Obesity: preventing and managing the global epidemic. Report on a WHO Consultation on Obesity, Geneva, 3–5 June, 1997. WHO/NUT/NCD/98.1. Technical Report Series Number 894. Geneva: World Health Organization, 2000.”

Point 4. Please include a perspective and possible application of these findings

Response 4. To reflect the review’s comment, we have added relavant discussion at line 339-340 in page 11, as described below

“We expected to provide the evidence on to identify the population at high risk of colorectal cancer via the results from the present study.”

Point 5. I suggest to include the following references for a more in depth discussion

Protein cell 2018;9(2):216-37 about the role of tricarboxylic acid in cancer

J Natl Cancer Inst 2010;102(14):1012-22 regarding the association of smoking with colon cancer

Response 5. To reflect the review’s suggestion, we have added the recommended references and the relavant discussion as described below,

Protein cell 2018;9(2):216-37, at line 202-203 in page 8

“Recent article has focused on targeting the citric acid cycle as the potential therapeutic strategy for cancer [18].”

J Natl Cancer Inst 2010;102(14):1012-22, at line 235-237 in page 8-9

“A prospective study defined the subtype of colorectal cancer based on microsatellite instability, CpG island methylator, and BRAF mutation and found that the association between smoking and colorectal cancer differs according to the molecular subtype of colorectal cancer among women [28].”

Point 6. Since it concerns metabolic cancer pathways, I suggest to include on line 61-62 page 2 the following about the activation of HIF and its function:

Oncotarget 2019;10(66):7071-79

Curr Molec Med 2018;18(6):343-51

Response 6. To reflect the review’s comment, we have added the suggested literature and the relavant discussion at line 60-62 in page 2, as described below

“An in vitro study reported a significant association between the intermediates of the citric acid cycle and the regulation of hypoxia-inducible factor (HIF), which is a transcription factor for angiogenesis, glucose utilization, and apoptosis [7, 8, 9].”

Reviewer 2 Report

The study by Cho and colleagues reports the association between polymorphisms in the citric acid cycle gens and colorectal cancer in UK.

The rationale of the study is that an alteration of a central pathway involved on cell metabolism may impair energy balance and contribute to colorectal cancer development.

This hypothesis is intriguing and a few studies investigate this aspect.

However, even if a strength of this work is the large cohort of subjects involved, it still remains an association study, so that causality relationships remain to be exploited.

Limits of the study should be clearly stated.

Author Response

Point 1: The study by Cho and colleagues reports the association between polymorphisms in the citric acid cycle gens and colorectal cancer in UK. he rationale of the study is that an alteration of a central pathway involved on cell metabolism may impair energy balance and contribute to colorectal cancer development. his hypothesis is intriguing and a few studies investigate this aspect.

However, even if a strength of this work is the large cohort of subjects involved, it still remains an association study, so that causality relationships remain to be exploited. Limits of the study should be clearly stated.

Response 1: We appreciate the reviewer’s valuable comments, which are helpful for improving our work. We agree with the reviewer’s concern about the limitation on causality relationship. As reviewer mentioned, we thought that our findings based on large population provide a step toward cause of cancer incidence related to energy metabolism, but the causality still remains inconclusive. As commented by the review, these limitations were added in the revised manuscript at line 231-233 in page 8, as described below.

“Although we anticipated to provide clues to the etiology in cancer development related to energy metabolism through the results of the present study based on a large population, the causality still remains inconclusive.”

Reviewer 3 Report

Cancers: 920441

Genetic Variants of Citric Acid Cycle and Interactions with Obesity, Physical Activity and Energy Intake on Risk of Colorectal Cancer: Results from A Nested Case-Control Study in the UK Biobank

Sooyoung Cho, Nan Song, Ji-Yeob Choi, Aesun Shin

In this study, the authors have evaluated the association between genetic variants of the mitochondrial citric acid cycle and colorectal cancer. The rationale behind this study is genetic polymorphisms in mitochondria, plays a central role in energy metabolism. 3,523 colorectal cancer cases and 10,522 matched controls from the UK Biobank study were analyzed. SUCLG2 gene rs35494829 SNP and CRC were significantly associated. significant SNP-SNP interactions among citric acid cycle SNPs in colorectal cancer were identified. The study primarily focusses on cancer metabolism.

PROS

  • The paper is well written, and the experiments are designed to support the hypothesis and conclusions.
  • The number of normal and disease tissue cohort makes it a significant study.
  • The interaction between SNPs in the gene encoding components of the citric acid cycle and contributors of energy balance on colorectal cancer risk was investigated. Authors have utilized their statistical strength in this analysis, how much it is relevant is not sure.
  • Access to patient follow up data is a big plus for the analysis.

CONS

Major

  • The rationale on selecting the SNPs for this study is not clear.
  • How the SNP chosen are relevant citric acid cycle marker, and they had a phenotype, need to be specified.

Minor

The statistics mentioned is of 2018, need be updated to 2020

Author Response

Response to Reviewer 3 Comments

 Point 1: The rationale on selecting the SNPs for this study is not clear. How the SNP chosen are relevant citric acid cycle marker, and they had a phenotype, need to be specified.

Response 1: We appreciate the reviewer’s valuable comments, which are helpful for improving our manuscript. As reviewer mentioned, we have added reference and the relevant discussion in “4.5 Marker selection” section at line 300-304 in page 10, as stated follow.:

“The MitoProteome database (available at http://www. mitoproteome.org/) was used to select the genes contributing to the citric acid cycle [36]. We find the citric acid cycle gene based on Kyoto encyclopedia of genes and genomes (KEGG) [37]using the keyword of "Citrate cycle (TCA cycle)" and 27 autosomal genes were extracted. Then, SNPs within the 27 genes related to the TCA cycle were found using the dbSNP database [38].”

Round 2

Reviewer 2 Report

The authors have now toned down conclusions regarding they study. Since the topic is under investigated, the work may be of interest for readers.